# A Graph Theoretic Additive Approximation of Optimal Transport

**Nathaniel Lahn**
Department of Computer Science
Virginia Tech
Blacksburg, VA 24061
lahnn@vt.edu

**Deepika Mulchandani**
Virginia Tech
Blacksburg, VA 24061
deepikak@vt.edu

**Sharath Raghvendra**
Virginia Tech
Blacksburg, VA 24061
sharathr@vt.edu

## Abstract

Transportation cost is an attractive similarity measure between probability distributions due to its many useful theoretical properties. However, solving optimal transport exactly can be prohibitively expensive. Therefore, there has been significant effort towards the design of scalable approximation algorithms. Previous combinatorial results [Sharathkumar, Agarwal STOC '12, Agarwal, Sharathkumar STOC '14] have focused primarily on the design of near-linear time multiplicative approximation algorithms. There has also been an effort to design approximate solutions with additive errors [Cuturi NIPS '13, Altschuler *et al.* NIPS '17, Dvurechensky *et al.* ICML '18, Quanrud, SOSA '19] within a time bound that is linear in the size of the cost matrix and polynomial in $C/\delta$; here $C$ is the largest value in the cost matrix and $\delta$ is the additive error. We present an adaptation of the classical graph algorithm of Gabow and Tarjan and provide a novel analysis of this algorithm that bounds its execution time by $\mathcal{O}(\frac{n^2 C}{\delta} + \frac{nC^2}{\delta^2})$. Our algorithm is extremely simple and executes, for an arbitrarily small constant $\varepsilon$, only $\lfloor \frac{2C}{(1-\varepsilon)\delta} \rfloor + 1$ iterations, where each iteration consists only of a Dijkstra-type search followed by a depth-first search. We also provide empirical results that suggest our algorithm is competitive with respect to a sequential implementation of the Sinkhorn algorithm in execution time. Moreover, our algorithm quickly computes a solution for very small values of $\delta$ whereas Sinkhorn algorithm slows down due to numerical instability.

## 1 Introduction

Transportation cost has been successfully used as a measure of similarity between data sets such as point clouds, probability distributions, and images. Originally studied in operations research, the transportation problem is a fundamental problem where we are given a set $A$ of 'demand' nodes and a set $B$ of 'supply' nodes with a non-negative demand of $d_a$ at node $a \in A$ and a non-negative supply $s_b$ at node $b \in B$. Let $G(A, B)$ be a complete bipartite graph on $A, B$ with $n = |A| + |B|$ where $c(a, b) \geq 0$ denotes the cost of transporting one unit of supply from $b$ to $a$; we assume that $C$ is the largest cost of any edge in the graph. We assume that the cost function is symmetric, i.e., $c(a, b) = c(b, a)$. Due to symmetry in costs, without loss of generality, we will assume throughout that the total supply is at most the total demand. Let $U = \sum_{b \in B} s_b$. A *transport plan* is a function $\sigma : A \times B \to \mathbb{R}_{\geq 0}$ that assigns a non-negative value $\sigma(a, b)$ to every edge $(a, b)$ with the constraints

that the total supply coming into any node $a \in A$ is at most $d_a$, i.e., $\sum_{b \in B} \sigma(a,b) \leq d_a$ and the total supply leaving a node $b \in B$ is at most $s_b$, i.e., $\sum_{a \in A} \sigma(a,b) \leq s_b$. A *maximum transport plan* is one where, for every $b \in B$, $\sum_{a \in A} \sigma(a,b) = s_b$, i.e., every available supply is transported. The cost incurred by any transport plan $\sigma$, denoted by $w(\sigma)$ is $\sum_{(a,b) \in A \times B} \sigma(a,b)c(a,b)$. In the transportation problem, we wish to compute the minimum-cost maximum transport plan.

There are many well-known special versions of the transportation problem. For instance, when all the demands and supplies are positive integers, the problem is the Hitchcock-Koopmans transportation problem. When the demand or supply at every node is 1, the problem becomes the *assignment problem*. When $A$ and $B$ are discrete probability distributions where each node has an associated probability, the total demand (resp. supply) will equal to 1, i.e., $U = 1$. This is the problem of computing *optimal transport* distance between two distributions. When the cost of transporting between nodes is a metric, the optimal transport cost is also the Earth Mover's distance (EMD). If instead, the costs between two nodes is the $p$-th power of some metric cost with $p > 1$, the optimal transport cost is also known as the $p$-Wasserstein's distance. These special instances are of significant theoretical interest [1, 17, 20, 24, 28, 29] and also have numerous applications in operations research, machine learning, statistics, and computer vision [5, 6, 7, 10, 13, 27].

**Related work:** There are several combinatorial algorithms for the transportation problem. The classical Hungarian method computes an optimal solution for the assignment problem by using linear programming duality in $\mathcal{O}(n^3)$ time [18]. In a seminal paper, Gabow and Tarjan applied the cost scaling paradigm and obtained an $\mathcal{O}(n^{2.5} \log(nC))$ time algorithm for the assignment problem [14]. They extended their algorithm to the transportation problem with an execution time of $\mathcal{O}((n^2\sqrt{U} + U \log U) \log(nC))$; their algorithm requires the demands, supplies and edge costs to be integers. For the optimal transport problem, scaling the demands and supplies to integers will cause $U$ to be $\Omega(n)$. Therefore, the execution time of the GT-Algorithm will be $\Omega(n^{2.5})$. Alternatively, one could use the demand scaling paradigm to obtain an execution time of $\mathcal{O}(n^3 \log U)$ [12]. For integer supplies and demands, the problem of computing a minimum-cost maximum transport plan can be reduced to the problem of computing a minimum-cost maximum flow, and applying the result of [21] gives a $\tilde{\mathcal{O}}(n^{2.5}\text{poly} \log(U))^1$ time algorithm.

We would also like to note that there is an $\mathcal{O}(n^\omega C)$ time algorithm to compute an optimal solution for the assignment problem [23]; here, $\omega$ is the exponent of matrix multiplication time complexity. With the exception of this algorithm, much of the existing work for exact solutions have focused on design of algorithms that have an execution time polynomial in $n, \log U$, and $\log C$. All existing exact solutions, however, are quite slow in practice. This has shifted focus towards the design of approximation algorithms.

Fundamentally, there are two types of approximate transport plans that have been considered. We refer to them as $\delta$-approximate and $\delta$-close transport plans and describe them next. Suppose $\sigma^*$ is a maximum transport plan with the smallest cost. A $\delta$-approximate transport plan is one whose cost is within $(1 + \delta)w(\sigma^*)$, whereas a transport plan $\sigma$ is $\delta$-close if its cost is at most an additive value $U\delta$ larger than the optimal, i.e., $w(\sigma) \leq w(\sigma^*) + U\delta$. Note that, for discrete probability distributions $U = 1$, and, therefore, a $\delta$-close solution is within an additive error of $\delta$ from the optimal.

For metric and geometric costs, there are several algorithms to compute $\delta$-approximate transport plans that execute in time near-linear in the input size and polynomial in $(\log n, \log U, \log C)$. For instance, $\delta$-approximate transport plans for the $d$-dimensional Euclidean assignment problem and the Euclidean transportation problem can be computed in $n(\log n/\delta)^{\mathcal{O}(d)} \log U$ time [17, 28]. For metric costs, one can compute a $\delta$-approximate transport plan in $\tilde{\mathcal{O}}(n^2)$ time [1, 30]. There are no known $\delta$-approximation algorithms that execute in near-linear time when the costs are arbitrary.

There are several algorithms that return $\delta$-close transport plan for any arbitrary cost function. Among these, the best algorithms take $\tilde{\mathcal{O}}(n^2(C/\delta))$ time. In fact, Blanchet *et al.* [8] showed that any algorithm with a better dependence on $C/\delta$ can be used to compute a maximum cardinality matching in any arbitrary bipartite graph in $o(n^{5/2})$ time. Therefore, design of improved algorithms with sub-linear dependence on $(C/\delta)$ seems extremely challenging. See Table 1 for a summary of various results for computing a $\delta$-close transport plan. Note that all previous results have one or more factors of

Table 1: A summary of existing algorithms for computing a $\delta$-close transport plan.

| Algorithm | Time Complexity |
|---|---|
| **Altschuler _et al._, '17** | $\tilde{\mathcal{O}}(n^2(C/\delta)^3)$ [3] |
| **Dvurechensky _et al._, '18** | $\tilde{\mathcal{O}}(\min(n^{9/4}\sqrt{C}/\delta, n^2C/\delta^2)$ [11] [2] |
| **Lin _et al._, '19** | $\tilde{\mathcal{O}}(\min(n^2C\sqrt{\gamma}/\delta, n^2(C/\delta)^2)$ [22] [2] |
| **Quanrud, '19** | $\tilde{\mathcal{O}}(n^2C/\delta)$ [25] |
| **Blanchet _et al._, '19** | $\tilde{\mathcal{O}}(n^2C/\delta)$ [8] |
| **Our Result** | $\mathcal{O}(n^2C/\delta + n(C/\delta)^2)$ |

$\log n$ in their execution time. While some of these poly-logarithmic factors are artifacts of worst-case analyses of the algorithms, one cannot avoid them all-together in any practical implementation.

Due to this, only a small fraction of these results have reasonable implementation that also perform well in practical settings. We would like to highlight the results of Cuturi [9], Altschuler _et al._ [3], and, Dvurechensky _et al._ [11], all of which are based on the Sinkhorn projection technique. All these implementations, however, suffer from a significant increase in running time and numerical instability for smaller values of $\delta$. These algorithms are practical only when $\delta$ is moderately large.

In an effort to design scalable solutions, researchers have explored several avenues. For instance, there has been an effort to design parallel algorithms [15]. Another avenue for speeding algorithms is to exploit the cost structure. For instance, Sinkhorn based algorithms can exploit the structure of squared Euclidean distances leading to a $\mathcal{O}(n(C/\delta)^d \log^d n)$ time algorithm that produces a $\delta$-close transport plan [4].

**Our results and approach:** We present a deterministic primal-dual algorithm to compute a $\delta$-close solution in $\mathcal{O}(n^2(C/\delta) + n(C/\delta)^2)$ time; note that $n^2(C/\delta)$ is the dominant term in the execution time provided $C/\delta$ is $\mathcal{O}(n)$. Our algorithm is an adaptation of a single scale of Gabow and Tarjan's scaling algorithm for the transportation problem. Our key contribution is a diameter-sensitive analysis of this algorithm. The dominant term in the execution time is linear in the size of the cost-matrix and linear in $(C/\delta)$. The previous results that achieve such a bound are randomized and have additional logarithmic factors [8, 25], whereas our algorithm does not have any logarithmic factors and is deterministic. Furthermore, we can also exploit the cost structure to improve the execution time of our algorithm to $\tilde{\mathcal{O}}(n(C/\delta)^2)$ for several geometric costs.

We transform our problem to one with integer demands and supplies in $\mathcal{O}(n^2)$ time (Section 1.1). Given the transformed demands and supplies, our algorithm (in Section 2) scales down the cost of every edge $(a, b)$ to $\lfloor \frac{2c(a,b)}{(1-\varepsilon)\delta} \rfloor$ for an arbitrarily small constant $0 < \varepsilon < 1$, and executes, at most $\lfloor 2C/((1-\varepsilon)\delta) \rfloor + 1$ phases. Within each phase, the algorithm executes two steps. The first step (also called the Hungarian Search) executes Dijkstra's algorithm $(\mathcal{O}(n^2))$ and adjusts the weights corresponding to a dual linear program to find an augmenting path consisting of zero slack edges. The second step executes DFS from every node with an excess supply and finds augmenting paths of zero slack edges to route these supplies. The time taken by this step is $\mathcal{O}(n^2)$ for the search and an additional time proportional to the sum of the lengths of all the augmenting paths found by the algorithm. We bound this total length of paths found during the course of the algorithm by $\mathcal{O}(n/\varepsilon(1-\varepsilon)(C/\delta)^2)$.

**Comparison with Gabow-Tarjan:** Our algorithm can be seen as executing a single scale of Gabow and Tarjan's algorithm for carefully scaled integer demand, integer supply, and integer cost functions. Let $\mathcal{U}$ be the total integer supply. Our analysis differs from Gabow and Tarjan's analysis in the following ways. Gabow and Tarjan's algorithm computes an optimal solution only when the total supply, i.e., $\mathcal{U}$ is equal to total demand. In fact, there has been substantial effort in extending it to the unbalanced case [26]. Our transformation to integer demands and supply in Section 1.1 makes the problem inherently unbalanced. However, we identify the fact that the difficulty with unbalanced demand and supply exists only when the algorithm executes multiple scales. We provide a proof that our algorithm works for the unbalanced case (see Lemma 2.1). To bound the number of phases by

$\mathcal{O}(\sqrt{\mathcal{U}})$ and the length of the augmenting paths by $\mathcal{O}(\mathcal{U} \log \mathcal{U})$, Gabow and Tarjan's proof requires the optimal solution to be of $\mathcal{O}(\mathcal{U})$ cost. We use a very different argument to bound the number of phases. Our proof (see Lemma 2.3 and Lemma 2.4) is direct and does not have any dependencies on the cost of the optimal solution.

**Experimental evaluations:** We contrast the empirical performance of our algorithm with the theoretical bounds presented in this paper and observe that the total length of the augmenting paths is substantially smaller than the worst-case bound of $n(C/\delta)^2$. We also compare our implementation with sequential implementations of Sinkhorn projection-based algorithms on real-world data. Our algorithm is competitive with respect to other methods for moderate and large values of $\delta$. Moreover, unlike Sinkhorn projection based approaches, our algorithm is numerically stable and executes efficiently for smaller values of $\delta$. We present these comparisons in Section 3.

**Extensions:** In Section 4, we discuss a faster implementation of our algorithm using a dynamic weighted nearest neighbor data structure with an execution time of $\tilde{\mathcal{O}}(n(C/\delta)\Phi(n))$. Here, $\Phi(n)$ is the query and update time of this data structure. As consequences, we obtain an $\tilde{\mathcal{O}}(n(C/\delta))$ time algorithm to compute $\delta$-close optimal transport for several settings including when $A, B \subset \mathbb{R}^2$ and the costs are Euclidean distances and squared-Euclidean distances.

## 1.1 Scaling demands and supplies

In this section, we transform the demands and supplies to integer demands and supplies. By doing so, we are able to apply the traditional framework of augmenting paths and find an approximate solution to the transformed problem in $\mathcal{O}(\frac{n^2 C}{\delta} + \frac{nC^2}{\delta^2})$ time. Finally, this solution is mapped to a feasible solution for the original demands and supplies. The total loss in accuracy in the cost due to this transformation is at most $\varepsilon U \delta$.

Let $0 < \varepsilon < 1$. Set $\alpha = \frac{2nC}{\varepsilon U \delta}$. Let $\mathcal{I}$ be the input instance for the transportation problem with each demand location $a \in A$ having a demand of $d_a$ and each supply location $b \in B$ having a supply of $s_b$. We create a new input instance $\mathcal{I}'$ by scaling the demand at each node $a \in A$ to $\overline{d}_a = \lceil d_a \alpha \rceil$ and scaling the supply at each node $b \in B$ to $\overline{s}_b = \lfloor s_b \alpha \rfloor$. Let the total supply be $\mathcal{U} = \sum_{b \in B} \overline{s}_b$. Since we scale the supplies by $\alpha$ and round them down, we have

$$\mathcal{U} = \sum_{b \in B} \overline{s}_b = \sum_{b \in B} \lfloor s_b \alpha \rfloor \leq \alpha \sum_{b \in B} s_b = \alpha U. \tag{1}$$

Recollect that for any input $\mathcal{I}$ to the transportation problem, the total supply is no more than the total demand. Since the new supplies are scaled by $\alpha$ and rounded down whereas the new demands are scaled by $\alpha$ and rounded up, the total supplies in $\mathcal{I}'$ remains no more than the total demand. Let $\sigma'$ be any feasible maximum transport plan for $\mathcal{I}'$. Now consider a transport plan $\sigma$ that sets, for each edge $(a,b)$, $\sigma(a,b) = \sigma'(a,b)/\alpha$. As described below, the transport plan $\sigma$ is not necessarily feasible or maximum for $\mathcal{I}$.

(i) $\sigma$ is not necessarily a maximum transport plan for $\mathcal{I}$ since the total supplies transported out of any node $b \in B$ is $\sum_{a \in A} \sigma(a,b) = \sum_{a \in A} \sigma'(a,b)/\alpha = \lfloor \alpha s_b \rfloor/\alpha \leq s_b$. Note that the excess supply remaining at any node $b \in B$ is $\kappa_b = s_b - \lfloor \alpha s_b \rfloor/\alpha \leq 1/\alpha$.

(ii) $\sigma$ is not a feasible plan for $\mathcal{I}$ since the total demand met at any node $a \in A$ can be more than $d_a$, i.e., $\sum_{b \in B} \sigma(a,b) = \sum_{b \in B} \sigma'(a,b)/\alpha = \overline{d}_a/\alpha = \lceil \alpha d_a \rceil/\alpha \geq d_a$. Note that the excess supply that reaches node $a \in A$, $\kappa_a \leq \lceil \alpha d_a \rceil/\alpha - d_a \leq \frac{\alpha d_a + 1}{\alpha} - d_a = 1/\alpha$.

The cost of $\sigma$, $w(\sigma) = w(\sigma')/\alpha$. We can convert $\sigma$ to a feasible and maximum transport plan for $\mathcal{I}$ in two steps.

First, one can convert $\sigma$ to a feasible solution. The excess supply $\kappa_a$ that reaches a demand node $a \in A$ can be removed by iteratively picking an arbitrary edge incident on $a$, say the edge $(a,b)$, and reducing $\sigma(a,b)$ as well as $\kappa_a$ by $\min\{\kappa_a, \sigma(a,b)\}$. This iterative process is applied until $\kappa_a$ reduces to 0. This step is also repeated at every demand node $a \in A$ with an $\kappa_a > 0$. The total excess supply pushed back will increase the leftover supply at the supply nodes by $\sum_{a \in A} \kappa_a \leq n/\alpha$. Combined with the left-over supply from (i), the total remaining supply in $\sigma$ is at most $2n/\alpha$. $\sigma$ is now a feasible transportation plan with an excess supply of at most $2n/\alpha$. Since the supplies transported along edges only reduce, the cost $w(\sigma) \leq w(\sigma')/\alpha$.

Second, to convert this feasible plan $\sigma$ to a maximum transport plan, one can simply match the remaining $2n/\alpha$ supplies arbitrarily to leftover demands at a cost of at most $C$ per unit of supply. The cost of this new transport plan increases by at most $2nC/\alpha$ and so,

$$w(\sigma) \le w(\sigma')/\alpha + \frac{2nC}{\alpha} \le w(\sigma')/\alpha + \varepsilon U\delta. \qquad (2)$$

Recollect that $\sigma^*$ is the optimal solution for $\mathcal{I}$. Let $\sigma'_{\mathrm{OPT}}$ be the optimal solution for input instance $\mathcal{I}'$. In Lemma 1.1 (proof of which is in the supplement), we show that $w(\sigma'_{\mathrm{OPT}}) \le \alpha w(\sigma^*)$. In the Section 2, we show how to construct a transport plan $\sigma'$ with a cost $w(\sigma') \le w(\sigma'_{\mathrm{OPT}}) + (1-\varepsilon)\mathcal{U}\delta$, which from Lemma 1.1, can be rewritten as $w(\sigma') \le \alpha w(\sigma^*) + (1-\varepsilon)\mathcal{U}\delta$. By combining this with equations (1) and (2), the solution produced by our algorithm is $w(\sigma) \le w(\sigma^*) + (1-\varepsilon)\mathcal{U}\delta/\alpha + \varepsilon U\delta \le w(\sigma^*) + (1-\varepsilon)U\delta + \varepsilon U\delta = w(\sigma^*) + U\delta$.

**Lemma 1.1.** *Let $\alpha > 0$, be a parameter. Let $\mathcal{I}$ be the original instance of the transportation problem and let $\mathcal{I}'$ be an instance scaled by $\alpha$. Let $\sigma^*$ be the minimum-cost maximum transport plan for $\mathcal{I}$ and let $\sigma'_{\mathrm{OPT}}$ be an minimum-cost maximum transport plan for $\mathcal{I}'$. Then $w(\sigma'_{\mathrm{OPT}}) \le \alpha w(\sigma^*)$.*

## 2 Algorithm for scaled demands and supplies

The input $\mathcal{I}'$ consists of a set of demand nodes $A$ with demand of $\overline{d}_a$ for each node $a \in A$ and a set of supply nodes $B$ with supply of $\overline{s}_b$ for each node $b \in B$ along with the cost matrix as input. Let $\sigma'_{\mathrm{OPT}}$ be the optimal transportation plan for $\mathcal{I}'$. In this section, we present a variant of Gabow and Tarjan's algorithm that produces a plan $\sigma'$ with $w(\sigma') \le w(\sigma'_{\mathrm{OPT}}) + (1-\varepsilon)\delta\mathcal{U}$ in $\mathcal{O}(\frac{n^2 C}{(1-\varepsilon)\delta} + \frac{nC^2}{\varepsilon(1-\varepsilon)\delta^2})$ time. We obtain our result by setting $\varepsilon$ to be a constant such as $\varepsilon = 0.5$.

**Definitions and notations:** Let $\delta' = (1-\varepsilon)\delta$. We say that a vertex $a \in A$ (resp. $b \in B$) is *free* with respect to a transportation plan $\sigma$ if $\overline{d}_a - \sum_{b \in B} \sigma(a,b) > 0$ (resp. $\overline{s}_b - \sum_{a \in A} \sigma(a,b) > 0$). At any stage in our algorithm, we use $A_F$ (resp. $B_F$) to denote the set of free demand nodes (resp. supply nodes). Let $\overline{c}(a,b) = \lfloor 2c(a,b)/\delta' \rfloor$ be the scaled cost of any edge $(a,b)$. Recollect that $w(\sigma)$ is the cost of any transport plan $\sigma$ with respect to $c(\cdot,\cdot)$. Similarly, we use $\overline{w}(\sigma)$ to denote the cost of any transport plan with respect to the $\overline{c}(\cdot,\cdot)$.

This algorithm is based on a primal-dual approach. The algorithm, at all times, maintains a transport plan that satisfies the dual feasibility conditions. Given a transport plan $\sigma$ along with a dual weight $y(v)$ for every $v \in A \cup B$, we say that $\sigma, y(\cdot)$ is 1-feasible if, for any two nodes $a \in A$ and $b \in B$,

$$\begin{aligned} y(a) + y(b) &\le \overline{c}(a,b) + 1 & \text{if } \sigma(a,b) < \min\{s_b, d_a\} & \qquad (3) \\ y(a) + y(b) &\ge \overline{c}(a,b) & \text{if } \sigma(a,b) > 0. & \qquad (4) \end{aligned}$$

These feasibility conditions are identical to the one introduced by Gabow and Tarjan but for costs that are scaled by $2/\delta'$ and rounded down. We refer to a 1-feasible transport plan that is maximum as a 1-optimal transport plan. Note that Gabow-Tarjan's algorithm is defined for balanced transportation problem and so a maximum transport plan will also satisfy all demands. However, in our case there may still be unsatisfied demands. To handle them, we introduce the following additional condition. Consider any 1-optimal transport plan $\sigma$ such that for every demand node $a \in A$,

(C) The dual weight $y(a) \le 0$ and, if $a$ is a free demand node, then $y(a) = 0$.

In Lemma 2.1, we show that any 1-optimal transport plan $\sigma$ with dual weights $y(\cdot)$ satisfying (C) has the desired cost bound, i.e., $w(\sigma) \le w(\sigma'_{\mathrm{OPT}}) + \delta'\mathcal{U}$.

**Lemma 2.1.** *Let $\sigma$ along with dual weights $y(\cdot)$ be a 1-optimal transport plan that satisfies (C). Let $\sigma' = \sigma'_{\mathrm{OPT}}$ be a minimum cost maximum transport plan. Then, $w(\sigma) \le w(\sigma') + \delta'\mathcal{U}$.*

In the rest of this section, we describe an algorithm to compute a 1-optimal transport plan that satisfies (C). To assist in describing this algorithm, we introduce a few definitions.

For any 1-feasible transport plan $\sigma$, we construct a directed *residual graph* with the vertex set $A \cup B$ and denote it by $\overrightarrow{G}_\sigma$. The edge set of $\overrightarrow{G}_\sigma$ is defined as follows: For any $(a,b) \in A \times B$ if $\sigma(a,b) = 0$, we add an edge directed from $b$ to $a$ and set its *residual capacity* to be $\min\{\overline{d}_a, \overline{s}_b\}$. Otherwise, if $\sigma(a,b) = \min\{\overline{d}_a, \overline{s}_b\}$, we add an edge from $a$ to $b$ with a residual capacity of $\sigma(a,b)$. In all other

cases, i.e., $0 < \sigma(a,b) < \min\{\overline{d}_a, \overline{s}_b\}$, we add an edge from $a$ to $b$ with a residual capacity of $\sigma(a,b)$ and an edge from $b$ to $a$ with a residual capacity of $\min\{\overline{d}_a, \overline{s}_b\} - \sigma(a,b)$. Any edge of $\overrightarrow{G}_\sigma$ directed from $a \in A$ to $b \in B$ is called a *backward* edge and any edge directed from $b \in B$ to $a \in A$ is called a *forward* edge. We set the cost of any edge between $a$ and $b$ regardless of their direction to be $\overline{c}(a,b) = \lfloor 2c(a,b)/\delta' \rfloor$. Any directed path in the residual network starting from a free supply vertex to a free demand vertex is called an *augmenting path*. Note that the augmenting path alternates between forward and backward edges with the first and the last edge of the path being a forward edge. We can augment the supplies transported by $k \geq 1$ units along an augmenting path $P$ as follows. For every forward edge $(a,b)$ on the path $P$, we raise the flow $\sigma(a,b) \leftarrow \sigma(a,b) + k$. For every backward edge $(a,b)$ on the path $P$, we reduce the flow $\sigma(a,b) \leftarrow \sigma(a,b) - k$. We define slack on any edge between $a$ and $b$ in the residual network as

$$s(a,b) = \overline{c}(a,b) + 1 - y(a) - y(b) \qquad \text{if } (a,b) \text{ is a forward edge,} \qquad (5)$$
$$s(a,b) = y(a) + y(b) - \overline{c}(a,b) \qquad \text{if } (a,b) \text{ is a backward edge} \qquad (6)$$

Finally, we define any edge $(a,b)$ in $\overrightarrow{G}_\sigma$ as admissible if $s(a,b) = 0$. The *admissible graph* $\overrightarrow{\mathcal{A}}_\sigma$ is the subgraph of $\overrightarrow{G}_\sigma$ consisting of the admissible edges of the residual graph.

## 2.1 The algorithm

Initially $\sigma$ is a transport plan where, for every edge $(a,b) \in A \times B$, $\sigma(a,b) = 0$. We set the dual weights of every vertex $v \in A \cup B$ to 0, i.e., $y(v) = 0$. Note that $\sigma$ and $y(\cdot)$ together form a 1-feasible transportation plan. Our algorithm executes in *phases* and terminates when $\sigma$ becomes a maximum transport plan. Within each phase there are two *steps*. In the first step, the algorithm conducts a Hungarian Search and adjusts the dual weights so that there is at least one augmenting path of admissible edges. In the second step, the algorithm computes at least one augmenting path and updates $\sigma$ by augmenting it along all paths computed. At the end of the second step, we guarantee that there is no augmenting path of admissible edges. The details are presented next.

**First step (Hungarian Search):** To conduct a Hungarian Search, we add two additional vertices $s$ and $t$ to the residual network. We add edges directed from $s$ to every free supply node, i.e., nodes with $\sum_{a \in A} \sigma(a,b) < \overline{s}_b$. We add edges from every free demand vertex to $t$. All edges incident on $s$ and $t$ are given a weight 0. The weight of every other edge $(a,b)$ of the residual network is set to its slack $s(a,b)$ based on its direction. We refer to the residual graph with the additional two vertices as the *augmented residual network* and denote it by $\mathcal{G}_\sigma$. We execute Dijkstra's algorithm from $s$ in the augmented residual network $\mathcal{G}_\sigma$. For any vertex $v \in A \cup B$, let $\ell_v$ be the shortest path from $s$ to $v$ in $\mathcal{G}_\sigma$. Next, the algorithm performs a dual weight adjustment. For any vertex $v \in A \cup B$, if $\ell_v \geq \ell_t$, the dual weight of $v$ remains unchanged. Otherwise, if $\ell_v < \ell_t$, we update the dual weight as follows: **(U1):** If $v \in A$, we set $y(v) \leftarrow y(v) - \ell_t + \ell_v$, **(U2):** Otherwise, if $v \in B$, we set $y(v) \leftarrow y(v) + \ell_t - \ell_v$.

This completes the description of the first step of the algorithm. The dual updates guarantee that, at the end of this step, the transport plan $\sigma$ along with the updated dual weights remain 1-feasible and there is at least one augmenting path in the admissible graph.

**Second step (partial DFS):** Initially $\mathcal{A}$ is set to the admissible graph, i.e., $\mathcal{A} \leftarrow \overrightarrow{\mathcal{A}}_\sigma$. Let $X$ denote the set of free supply nodes in $\mathcal{A}$. The second step of the algorithm will iteratively initiate a DFS from each supply node of $X$ in the graph $\mathcal{A}$. We describe the procedure for one free supply node $b \in X$. During the execution of DFS from $b$, if a free demand node is visited, then an augmenting path $P$ is found, the DFS terminates immediately, and the algorithm deletes all edges visited by the DFS, except for the edges of $P$. The AUGMENT procedure augments $\sigma$ along $P$ and updates $X$ to denote the free supply nodes in $\mathcal{A}$. Otherwise, if the DFS ends without finding an augmenting path, then the algorithm deletes all vertices and edges that were visited by the DFS from $\mathcal{A}$ and updates $X$ to represent the free supply nodes remaining in $\mathcal{A}$. The second step ends when $X$ becomes empty.

**Augment procedure:** For any augmenting path $P$ starting at a free supply vertex $b \in B_F$ and ending at a free demand vertex $a \in A_F$, its *bottleneck edge set* is the set of all edges $(u,v)$ on $P$ with the smallest residual capacity. Let $bc(P)$ denote the capacity of any edge in the bottleneck edge set. The bottleneck capacity $r_P$ of $P$ is the smallest of the total remaining supply at $b$, the total remaining demand at $a$, and the residual capacity of its bottleneck edge, i.e., $r_P = \min\{\overline{s}_b - \sum_{a' \in A} \sigma(a',b), \overline{d}_a - \sum_{b' \in B} \sigma(a,b'), bc(P)\}$. The algorithm augments along $P$ by updating $\sigma$ as

follows. For every forward edge $(a', b')$, we set $\sigma(a', b') \leftarrow \sigma(a', b') + r_P$, and, for every backward edge $(a', b')$, $\sigma(a', b') \leftarrow \sigma(a', b') - r_P$. The algorithm then updates the residual network and the admissible graph to reflect the new transport plan.

**Invariants:** The following invariants (proofs of which are in the supplement) hold during the execution of the algorithm. **(I1):** The algorithm maintains a 1-feasible transport plan, and, **(I2)** In each phase, the partial DFS step computes at least one augmenting path. Furthermore, at the end of the partial DFS, there is no augmenting path in the admissible graph.

**Correctness:** From (I2), the algorithm augments, in each phase, the transport plan by at least one unit of supply. Therefore, when the algorithm terminates we have a 1-feasible (from (I1)) maximum transport plan, i.e., 1-optimal transport plan. Next, we show that any transport plan maintained by the algorithm will satisfy condition (C). For $v \in A$, initially $y(v) = 0$. In any phase, suppose $\ell_v < \ell_t$. Then, the Hungarian Search updates the dual weights using condition (U1) which reduces the dual weight of $v$. Therefore, $y(v) \leq 0$.

Next, we show that all free vertices of $A$ have a dual weight of $0$. The claim is true initially. During the course of the algorithm, any vertex $a \in A$ whose demand is met can no longer become free. Therefore, it is sufficient to argue that no free demand vertex experiences a dual adjustment. By construction, there is a directed edge from $v$ to $t$ with zero cost in $\mathcal{G}_\sigma$. Therefore, $\ell_t \leq \ell_v$ and the algorithm will not update the dual weight of $v$ during the phase. As a result the algorithm maintains $y(v) = 0$ for every free demand vertex and (C) holds.

When the algorithm terminates, we obtain a 1-optimal transport plan $\sigma$ which satisfies (C). From Lemma 2.1, it follows that $w(\sigma) \leq w(\sigma'_{\text{OPT}}) + \mathcal{U}\delta'$ as desired. The following lemma helps in achieving a diameter sensitive analysis of our algorithm.

**Lemma 2.2.** *The dual weight of any free supply node $v \in B_F$ is at most $\lfloor 2C/\delta' \rfloor + 1$.*

*Proof.* For the sake of contradiction, suppose the free supply node $v \in B_F$ has a dual weight $y(b) \geq \lfloor 2C/\delta' \rfloor + 2$. For any free demand node, say $a \in A_F$ has a dual weight $y(a) = 0$ (from (C)). Then, $y(a) + y(b) \geq \lfloor 2C/\delta' \rfloor + 2 \geq \bar{c}(a, b) + 2$, and the edge $(a, b)$ violates 1-feasibility condition (3) leading to a contradiction. $\square$

**Efficiency:** Let $\mathbb{P}_j$ be the set of all augmenting paths computed in phase $j$ and let $\mathbb{P}$ be the set of all augmenting paths computed by the algorithm across all phases. To bound the execution time of the algorithm, we bound, in Lemma 2.3, the total number of phases by $\lfloor 2C/\delta' \rfloor + 1$. Within each phase, the Hungarian search step executes a single Dijkstra search which takes $\mathcal{O}(n^2)$ time. To bound the time taken by the partial DFS step, observe that any edge visited by the DFS is deleted provided it does not lie on an augmenting path. Edges that lie on an augmenting path, however, can be visited again by another DFS within the same phase. Therefore, the total time taken by the partial DFS step in any phase $j$ is bounded by $\mathcal{O}(n^2 + \sum_{P \in \mathbb{P}_j} |P|)$; here $|P|$ is the number of edges on the augmenting path $P$. Across all $\mathcal{O}(C/\delta')$ phases, the total time taken is $\mathcal{O}((C/\delta')n^2 + \sum_{P \in \mathbb{P}} |P|)$. In Lemma 2.4, we bound the total length of the augmenting paths by $\mathcal{O}(\frac{n}{\varepsilon(1-\varepsilon)}(C/\delta)^2)$. Therefore, the total execution time of the algorithm is $\mathcal{O}(\frac{n^2 C}{(1-\varepsilon)\delta} + \frac{nC^2}{\varepsilon(1-\varepsilon)\delta^2})$.

**Lemma 2.3.** *The total number of phases in our algorithm is at most $\lfloor 2C/\delta' \rfloor + 1$.*

*Proof.* At the start of any phase, from (I2), there are no admissible augmenting paths. Therefore, any path from $s$ to $t$ in the augmented residual network $\mathcal{G}_\sigma$ will have a cost of at least 1, i.e., $\ell_t \geq 1$. During any phase, let $b \in B_F$ be any free supply vertex. Note that $b$ is also a free supply vertex in all prior phases. Since there is a direct edge from $s$ to $b$ with a cost of 0 in $\mathcal{A}$, $\ell_b = 0$. Since $\ell_t \geq 1$, from (U2), the dual weight of $b$ increases by at least 1. After $\lfloor 2C/\delta' \rfloor + 2$ phases, the dual weight of any free vertex will be at least $\lfloor 2C/\delta' \rfloor + 2$ which contradicts Lemma 2.2. $\square$

**Lemma 2.4.** *Let $\mathbb{P}$ be the set of all augmenting paths produced by the algorithm. Then $\sum_{P \in \mathbb{P}} |P| = \mathcal{O}(\frac{nC^2}{\varepsilon(1-\varepsilon)\delta^2})$; here $|P|$ is the number of edges on the path $P$.*

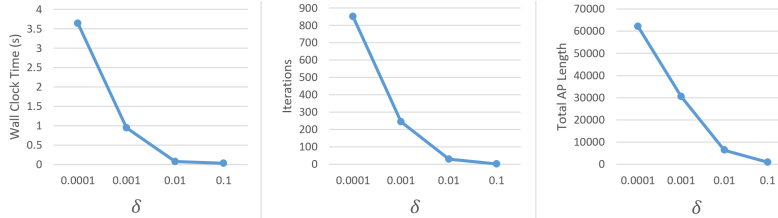

Figure 1: Efficiency statistics for our algorithm when executed on very small $\delta$ values.

# 3    Experimental Results

In this section, we investigate the practical performance of our algorithm. We test an implementation of our algorithm[3], written in Java, on discrete probability distributions derived from real-world image data. The testing code is written in MATLAB, and calls the compiled Java code. All tests are executed on computer with a 2.40 GHz Intel Dual Core i5 processor and 8GB of RAM, using a single computation thread. We implement a worst-case asymptotically optimal algorithm as presented in this paper and fix the value of $\varepsilon$ as $0.5$. We compare our implementation to existing implementations of the Sinkhorn, Greenkhorn[4], and APDAGD[5] algorithms, all of which are written in MATLAB. Unless otherwise stated, we set all parameters of these algorithms to values prescribed by the theoretical analysis presented in the respective papers.

All the tests are conducted on real-world data generated by randomly selecting pairs of images from the MNIST data set of handwritten digits. We set supplies and demands based on pixel intensities, and normalize such that the total supply and demand are both equal to $1$. The cost of an edge is assigned based on squared-Euclidean distance between pixel coordinates, and costs are scaled such that the maximum edge cost $C = 1$. The MNIST images are $28 \times 28$ pixels, implying each image has $784$ pixels.

In the first set of tests, we compare the empirical performance of our algorithm to its theoretical analysis from Section 2. We execute $100$ runs, where each run executes our algorithm on a randomly selected pair of MNIST images for $\delta \in [0.0001, 0.1]$. For each value of $\delta$, we record the wall-clock running time, iteration count, and total augmenting path length of our algorithm. These values, averaged over all runs, are plotted in Figure 1. We observe that the number of iterations is significantly less than the theoretical bound of roughly $\frac{2}{(1-\varepsilon)\delta} = 4/\delta$. We also observe that the total augmenting path length is significantly smaller than the worst case bound of $n/\delta^2$ for even for the very small $\delta$ value of $0.0001$. This is because of two reasons. First, the inequality (8) (Proof of Lemma 2.4 in the supplement) used in bounding the augmenting path length is rarely tight; most augmenting paths have large slack with respect to this inequality. Second, to obtain the worst case bound, we assume that only one unit of flow is pushed through each augmenting path (see Proof of Lemma 2.4). However, in our experiments augmenting paths increased flow by a larger value whenever possible. As a result, we noticed that the total time taken by augmentations, even for the smallest value of $\delta$, was negligible and $\leq 2\%$ of the execution time.

Next, we compare the number of iterations executed by our algorithm with the number of iterations executed by the Sinkhorn, Greenkhorn, and APDAGD algorithms. Here, by 'iteration' we refer to the the logical division of each algorithm into portions that take $\mathcal{O}(n^2)$ time. For example, an iteration of the Greenkhorn algorithm corresponds to $n$ row/column updates. An iteration for our algorithm corresponds to a single phase. We execute 10 runs, where each run selects a random pair of MNIST images and executes all four algorithms using $\delta$ values in the range $[0.025, 0.2]$.

Figure 2(a) depicts the average number of iterations for each algorithm. For the Sinkhorn, Greenkhorn and APDAGD algorithms, we see a significant increase in iterations as we choose smaller values of $\delta$. We believe this is because of numerical instability associated with these algorithms. Unlike these algorithms, our algorithm runs fairly quickly for very small values of $\delta$ (see Figure 1).

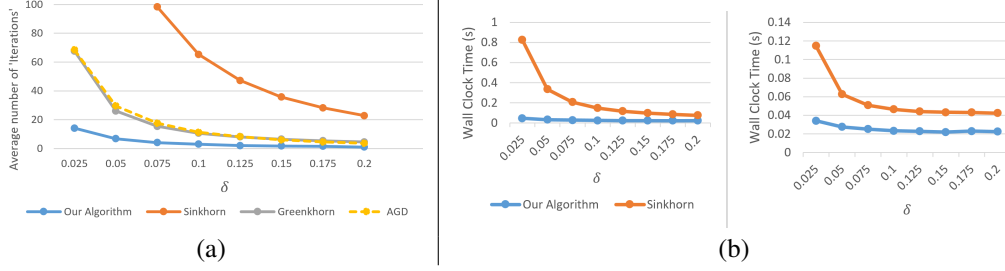

Figure 2: (a) A comparison of the number of iterations executed by various algorithms for moderate values of $\delta$; (b) A comparison of our algorithm with the Sinkhorn algorithm using several $\delta$ values. We compare running times when both algorithms receive $\delta$ as input (left) and compare the running times when Sinkhorn receives $5\delta$ and our algorithm receives $\delta$ (right).

We repeat the test from Figure 2(a), and compare the average time taken to produce the result. The code for Greenkhorn and APDAGD are not optimized for actual running time, so we restrict to comparing our algorithm with the Sinkhorn algorithm, and record average wall-clock running time. The results of executing 100 runs are plotted on the left in Figure 2(b). Under these conditions, we observe that the time taken by our algorithm is significantly less than that taken by the Sinkhorn algorithm. The cost of the solution produced by our algorithm, although within the error parameter $\delta$, was not always better than that of the cost of the solution produced by Sinkhorn. We repeat the same experimental setup, except with Sinkhorn receiving an error parameter of $5\delta$ while our algorithm continues to receive $\delta$. Our algorithm continues to have a better average execution time than Sinkhorn (plot on the right in Figure 2(b)). Moreover, we also observe that cost of solution produced by Sinkhorn is higher than the solution produced by our algorithm for every run across all values of $\delta$. In other words, for this experiment, our algorithm executed faster and produced a better quality solution than the Sinkhorn algorithm.

## 4 Extensions and Conclusion

We presented an $\mathcal{O}(n^2(C/\delta) + n(C/\delta)^2)$ time algorithm to compute a $\delta$-close approximation of the optimal transport. Our algorithm is an execution of a single scale of Gabow and Tarjan's algorithm for appropriately scaled integer demands, supplies and costs. Our key technical contribution is a diameter sensitive analysis of the execution time of this algorithm.

In [29, Section 3.1, 3.2], it has been shown that the first and the second steps of our algorithm, i.e., Hungarian search and partial DFS, can be executed on a residual graph with costs $\overline{c}(\cdot, \cdot)$ in $\mathcal{O}(n\Phi(n)\log^2 n)$ and $\mathcal{O}(n\Phi(n))$ time respectively; here $\Phi(n)$ is the query and update time of a dynamic weighted nearest neighbor data structure with respect to the cost function $c(\cdot, \cdot)$. Unlike in [29] where the costs, after scaling down by a factor $\delta'$, are rounded up, $\overline{c}(\cdot, \cdot)$ is obtained by scaling down $c(\cdot, \cdot)$ by $\delta'$, and is rounded down. This, however, does not affect the critical lemma [29, Lemma 3.2] and the result continues to hold. Several distances including the Euclidean distance and the squared-Euclidean distance admit such a dynamic weighted nearest neighbor data structure for planar point sets with poly-logarithmic query and update time [2, 16]. Therefore, we immediately obtain a $\tilde{\mathcal{O}}(n(C/\delta)^2)$ time algorithm to compute a $\delta$-close approximation of the optimal transport for such distances. In [1, Section 4 (i)–(iii)], a similar link is made between an approximate nearest neighbor (ANN) data structure and a relative approximation algorithm.

The Sinkhorn algorithm scales well in parallel settings because the row and column update operations within each iteration can be easily parallelized. In the first step, our algorithm uses Dijkstra's method that is inherently sequential. However, there is an alternate implementation of a single scale of Gabow and Tarjan's algorithm that does not require Dijkstra's search (see [19, Section 2.1]) and may be more amenable to parallel implementation. We conclude with the following open questions:

- Can we design a combinatorial algorithm that uses approximate nearest neighbor data structure and produces a $\delta$-close transport plan for geometric costs in $\tilde{\mathcal{O}}(n(C/\delta)^d)$ time?
- Can we design a parallel combinatorial approximation algorithm that produces a $\delta$-close optimal transport for arbitrary costs?

**Acknowledgements:** Research presented in this paper was funded by NSF CCF-1909171. We would like to thank the anonymous reviewers for their useful feedback. All authors contributed equally to this research.

## Footnotes

[1]We use the notation $\tilde{\mathcal{O}}$ to suppress additional logarithmic terms in $n$.

[2]These results have an additional data dependent parameter in the running time. For the result in Lin _et al._, $\gamma = \mathcal{O}(n)$ is this parameter.

[3]Our implementation is available at https://github.com/nathaniellahn/CombinatorialOptimalTransport.

[4]Sinkhorn/Greenkhorn implementations retrieved from https://github.com/JasonAltschuler/OptimalTransportNIPS17.

[5]APDAGD implementation retrieved from https://github.com/chervud/AGD-vs-Sinkhorn.

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
