[Supplementary Material · Optimal_Transport_NeurIPS_final_supplement.pdf]

# Supplement to "A Graph Theoretic Additive Approximation of Optimal Transport"

**Nathaniel Lahn**
Department of Computer Science
Virginia Tech
Blacksburg, VA 24061
lahnn@vt.edu

**Deepika Mulchandani**
Virginia Tech
Blacksburg, VA 24061
deepikak@vt.edu

**Sharath Raghvendra**
Virginia Tech
Blacksburg, VA 24061
sharathr@vt.edu

## 1 Validity of the transport plan after augmentation

Note that the updated transport plan after each augmentation is valid. Augmenting along $P$ by the residual capacity $r_P$ will maintain a feasible flow and so, for any edge $(a, b) \in P$, $\sigma(a, b) \geq 0$. By our choice, $r_P$ is lower than the remaining demand at $a$ and the unused supply at $b$. Therefore, after augmentation, the supplies transported to $a$ (resp. from $b$) increases by $r_P$ and remains at most $\overline{d}_a$ (resp. $\overline{s}_b$).

For any other demand (resp. supply) node $a' \in A \cap P$ (resp. $b' \in B \cap P$) with $a' \neq a$ (resp. $b' \neq b$), the total supplies transported to $a'$ (resp. from $b'$) after the transport plan is updated remains unchanged. This is because $a'$ (resp. $b'$) has exactly one forward and one backward edge of $P$ incident on it. The increase in supply transported to $a'$ (resp. from $b'$) via the forward edge is canceled out by the decrease in supply transported along the backward edge.

## 2 Proof of Lemma 1.1

*Proof.* To prove our claim, it suffices to construct a maximum transport plan $\sigma'$ for $\mathcal{I}'$ such that $w(\sigma') \leq \alpha w(\sigma^*)$.

For any $(a, b) \in A \times B$, we define $\sigma'(a, b) = \alpha \sigma^*(a, b)$. Note that the transport plan $\sigma'$ is not necessarily valid for $\mathcal{I}'$. This is because the total flow of supplies from any supply node $b \in B$ exceeds $\overline{s}_b$ by

$$\kappa_b = \sum_{a \in A} \sigma'(a, b) - \overline{s}_b = \sum_{a \in A} \alpha \sigma^*(a, b) - \overline{s}_b = \alpha s_b - \lfloor \alpha s_b \rfloor.$$

The third equality follows from the fact that $\sigma^*$ is a maximum transport plan and so, $\sum_{a \in A} \sigma^*(a, b) = s_b$. However, $\sigma'$ satisfies the demand constraints. Specifically, for any demand node $a \in A$, combining the facts that $\sum_{b \in B} \sigma^*(a, b) \leq d_a$ and $\sigma'(a, b) = \alpha \sigma^*(a, b)$ we get,

$$\sum_{b \in B} \sigma'(a, b) \leq \alpha d_a \leq \lceil \alpha d_a \rceil = \overline{d}_a.$$

To make $\sigma'$ a valid maximum transportation plan, for every supply node $b \in B$, we iteratively choose an edge incident on $b$, say $(a, b)$ and reduce the flow of supplies along $(a, b)$ in $\sigma'(a, b)$ and the

excess supply $\kappa_b$, by $\min\{\sigma'(a,b), \kappa_b\}$. We continue this iterative process until $\kappa_b$ reduces to 0. By repeating this iterative process for every supply node, $\sigma'$ will satisfy the supply constraints with equality and is a maximum transport plan for $\mathcal{I}'$. Furthermore, since the supply transported in $\sigma'$ along the edge $(a,b)$ is at most $\alpha\sigma^*(a,b)$

$$w(\sigma') = \sum_{(a,b)\in A\times B} \sigma'(a,b)c(a,b) \le \sum_{(a,b)\in A\times B} \alpha\sigma^*(a,b)c(a,b) = \alpha w(\sigma^*).$$

□

## 3 Proof of Lemma 2.1

*Proof.* First, without loss of generality, we transform $\sigma$ and $\sigma'$ so that $\sigma$ remains a 1-optimal transport plan and $\sigma'$ remains the minimum-cost maximum transport plan. Furthermore, this transformation guarantees that the dual weights for the 1-optimal transport plan $\sigma$ are such that every edge $(a,b)$ for which the optimal transport plan has a positive flow, i.e., $\sigma'(a,b) > 0$, also satisfies (3). We present this transformation next. If the dual weights for an edge $(a,b)$ do not satisfy (3), then its flow $\sigma(a,b)$ is $\min\{s_b, d_a\}$. We reduce $d_a$ and $s_b$ by $\sigma'(a,b)$ and also reduce the flow on the edge $(a,b)$ in $\sigma'$ to be 0 and $\sigma$ to $\min\{s_b, d_a\} - \sigma'(a,b)$. The transformed $\sigma$ and $\sigma'$ continue to be 1-optimal transport plan and minimum cost maximum transport plan respectively for the new demands and supplies. Moreover, their difference in costs of $\sigma$ and $\sigma'$ does not change due to this transformation and we are guaranteed that if the edge $(a,b)$ has a positive flow with respect to $\sigma'$, i.e., $\sigma'(a,b) > 0$ then $(a,b)$ will satisfy (3). We present the rest of the proof assuming that $\sigma$ and $\sigma'$ are transformed.

The weight of $\sigma$ is

$$\begin{aligned} w(\sigma) &= \sum_{(a,b)\in A\times B} \sigma(a,b)c(a,b) \le \sum_{(a,b)\in A\times B} (\delta'/2)(\sigma(a,b)(\lfloor 2c(a,b)/\delta'\rfloor + 1)) \\ &\le (\delta'/2)\sum_{a\in A}(\sum_{b\in B}\sigma(a,b)y(a)) + (\delta'/2)\sum_{b\in B}(\sum_{a\in A}\sigma(a,b)y(b)) + \delta'\mathcal{U}/2. \end{aligned} \quad (1)$$

The last inequality follows from the fact that the dual weights for every edge $(a,b)$ with $\sigma(a,b) > 0$ satisfies (4). Note that, for any node $a \in A$, if $y(a) = 0$, then $\sum_{b\in B}\sigma(a,b)y(a) = \bar{d}_a y(a) = 0$. For every node $a \in A$ with $y(a) < 0$, by our assumption (C), $\sigma$ satisfies all the demands at $a$ and so, $\sum_{b\in B}\sigma(a,b)y(a) = \bar{d}_a y(a)$. Therefore, the first term in the RHS of (1) can be written as $\sum_{a\in A}\bar{d}_a y(a)$.

Since $\sigma$ is a maximum transport plan, the supplies available at each node $b \in B$ are completely transported by $\sigma$. Therefore, $\sum_{a\in A}\sigma(a,b)y(b) = \bar{s}_b y(b)$, and the second term in the RHS of (1) can be written as $\sum_{b\in B}\bar{s}_b y(b)$. Combined together,

$$w(\sigma) \le (\delta'/2)(\sum_{a\in A}\bar{d}_a y(a) + \sum_{b\in B}\bar{s}_b y(b) + \mathcal{U}). \quad (2)$$

The weight of the optimal transport plan $\sigma'$ can be written as

$$w(\sigma') = \sum_{(a,b)\in A\times B}\sigma'(a,b)c(a,b) \ge (\delta'/2)\sum_{(a,b)\in A\times B}\sigma(a,b)((\lfloor 2c(a,b)/\delta'\rfloor + 1) - 1).$$

Due to the initial transformation, every edge that has a positive flow in $\sigma'$ satisfies 1-feasibility condition (3). So, we can rewrite the above inequality as

$$w(\sigma') \ge (\delta'/2)\sum_{a\in A}(\sum_{b\in B}\sigma'(a,b)y(a)) + (\delta'/2)\sum_{b\in B}(\sum_{a\in A}\sigma'(a,b)y(b)) - \delta'\mathcal{U}/2. \quad (3)$$

For each demand node $a \in A$, the total flow of supply coming in to $a$ cannot exceed $\bar{d}_a$. Therefore,

$$\sum_{b\in B}\sigma'(a,b) \le \bar{d}_a.$$

Since $y(a) \leq 0$, we get

$$\sum_{b \in B} \sigma'(a,b)y(a) \geq \overline{d}_a y(a). \tag{4}$$

Since $\sigma'$ is a maximum transport plan, for each supply node $b \in B$, the total flow of supply going out of $b$ is exactly equal to $\overline{s}_b$. Therefore,

$$\sum_{a \in A} \sigma(a,b)y(b) = \overline{s}_b y(b). \tag{5}$$

Combining (4) and (5) with (3), we get

$$w(\sigma') \geq (\delta'/2) \sum_{a \in A} \overline{d}_a y(a) + (\delta'/2) \sum_{b \in B} \overline{s}_b y(b) - (\delta'\mathcal{U}/2). \tag{6}$$

Combining (6) with (2), we get $w(\sigma) \leq w(\sigma') + \delta'\mathcal{U}$. $\qquad\square$

## 4  Proof of Lemma 2.4

*Proof.* For any transportation plan $\sigma$, recollect that $\overline{w}(\sigma) = \sum_{(a,b) \in A \times B} \sigma(a,b)\overline{c}(a,b)$. For any augmenting path $P \in \mathbb{P}$, we let $P^{\uparrow}$ (resp. $P^{\downarrow}$) denote the set of forward (resp. backward) edges in $P$. We define the net-cost for any augmenting path $P \in \mathbb{P}$ from a free supply node $b$ to a free demand node $a$ as $\Phi(P) = \sum_{(a',b') \in P^{\uparrow}} (\overline{c}(a',b') + 1) - \sum_{(a',b')P^{\downarrow}} \overline{c}(a',b')$.

To bound the length of the augmenting paths, we provide two different bounds for net-cost. First, we bound the net-cost of $P$ by $\lfloor 2C/\delta' \rfloor + 1$.

$$\Phi(P) = \sum_{(a',b')P^{\uparrow}} (y(a') + y(b')) - \sum_{(a',b') \in P^{\downarrow}} (y(a') + y(b')) \tag{7}$$

$$= y(b) \leq \lfloor 2C/\delta' \rfloor + 1. \tag{8}$$

Equation (7) follows from the fact that $P$ is an augmenting path in the admissible graph and the slack on every edge of $P$ is zero. Equation (8) follows from Lemma 2.2 and the fact that every vertex except $a$ and $b$ has exactly one forward and one backward edge incident on it, and so their dual weights get canceled. Furthermore, when the augmenting path $P$ is found by the algorithm, $a$ is a free demand node, and, from (C), $y(a) = 0$.

Let $\sigma$ be the transport plan when $P$ was discovered by the algorithm. Let $\sigma'$ be the transport plan obtained after augmenting $\sigma$ along $P$ by $r_P$ units. Then,

$$r_P\Phi(P) = (\sum_{(a',b') \in P^{\uparrow}} r_P \cdot \overline{c}(a',b')) - (\sum_{(a',b') \in P^{\downarrow}} r_P \cdot \overline{c}(a',b')) + r_P\lceil |P|/2 \rceil \tag{9}$$

$$= \overline{w}(\sigma') - \overline{w}(\sigma) + r_P\lceil |P|/2 \rceil \tag{10}$$

(9) follows from the fact that there are exactly $\lceil |P|/2 \rceil$ forward edges in any augmenting path. (10) follows from the fact that $\sigma$ and $\sigma'$ differ in the flow assignment $\sigma(a',b')$ for every edge $(a',b') \in P$. In particular, for a forward edge $(a',b')$, $\sigma'(a',b') = \sigma(a',b') + r_P$ and for a backward edge $(a',b') \in P$, $\sigma'(a',b') = \sigma(a,b) - r_P$. Let $\overline{\sigma}$ be the 1-optimal transport plan returned by the algorithm. When we sum (9) across all augmenting paths computed by the algorithm, the cost of all intermediate transport plans computed by the algorithm cancel each other and we get $\sum_{P \in \mathbb{P}}(r_P\Phi(P)) = \overline{w}(\overline{\sigma}) + \sum_{P \in \mathbb{P}} r_P\lceil |P|/2 \rceil$,

$$\mathcal{U}(\lfloor 2C/\delta' \rfloor + 1) \geq \sum_{P \in \mathbb{P}} \lceil |P|/2 \rceil. \tag{11}$$

(11) follows from the facts that $\overline{w}(\overline{\sigma}) \geq 0$, $r_P \geq 1$, and $\sum_{P \in \mathbb{P}} r_P \leq \mathcal{U}$, i.e., the total flow pushed along all augmenting paths is at most $\mathcal{U}$. From the fact that $\mathcal{U} \leq \alpha U = \frac{2nC}{\varepsilon\delta}$, we get $\sum_{P \in \mathbb{P}} |P| = \mathcal{O}(\frac{nC}{\varepsilon\delta} \frac{C}{(1-\varepsilon)\delta})$. $\qquad\square$

## 4.1 Proof of Invariants

The following two invariants are maintained by the algorithm. The proof of both are classical. For the sake of completion, we present the proofs here. **(I1):** The algorithm maintains a 1-feasible transport plan, and, **(I2)** In each phase, the partial DFS step computes at least one augmenting path. Furthermore, at the end of this step, there is no augmenting path in the admissible graph.

**Proof of (I1):** We show that the dual updates conducted by Hungarian search do not violate (I1), i.e., every forward edges satisfies (3) and every backward edge satisfies (4). For any directed edge $(u, v)$ in the residual graph, and from the shortest path property,

$$\ell_u + s(u, v) \geq \ell_v. \tag{12}$$

There are four possibilities: (i) $\ell_u < \ell_t$ and $\ell_v < \ell_t$, (ii) $\ell_u \geq \ell_t$ and $\ell_v < \ell_t$, (iii) $\ell_u < \ell_t$ and $\ell_v \geq \ell_t$ or (iv) $\ell_u \geq \ell_t$ and $\ell_v \geq \ell_t$. Hungarian search does not update the dual weights for $u$ and $v$ in case (iv) and so any such forward (resp. backward) edge continues to satisfy (3) (resp. (4)). We present the proof for the other three cases.

For case (i), if $(u, v)$ is a forward (resp. backward) edge then $u \in B$, $v \in A$ (resp. $u \in A$, $v \in B$). The updated dual weights $\tilde{y}(u) = y(u) + \ell_t - \ell_u$ (resp. $\tilde{y}(u) = y(u) - \ell_t + \ell_u$) and $\tilde{y}(v) = y(v) - \ell_t + \ell_v$ (resp. $\tilde{y}(v) = y(v) + \ell_t - \ell_v$) and the updated feasibility condition is $\tilde{y}(u) + \tilde{y}(v) = y(u) + y(v) + \ell_v - \ell_u$ (resp. $\tilde{y}(u) + \tilde{y}(v) = y(u) + y(v) - \ell_v + \ell_u$). From (12), $\tilde{y}(u) + \tilde{y}(v) \leq y(v) + y(v) + s(u, v) = \bar{c}(u, v) + 1$ (resp. $\tilde{y}(u) + \tilde{y}(v) \geq y(v) + y(v) - s(u, v) = \bar{c}(u, v)$) . The last equality follows from the definition of slack for a forward (resp. backward) edge.

For case (ii), if $(u, v)$ is a forward (resp. backward) edge then $u \in B$, $v \in A$ (resp. $u \in A$, $v \in B$). The updated dual weights are $\tilde{y}(u) = y(u)$ and $\tilde{y}(v) = y(v) - \ell_t + \ell_v$ (resp. $\tilde{y}(v) = y(v) + \ell_t - \ell_v$). Since the dual weight of $v$ reduces (resp. increases) and that of $u$ remains unchanged, their sum only reduces (resp. increases) and $\tilde{y}(u) + \tilde{y}(v) = y(u) + y(v) + \ell_v - \ell_t \leq \bar{c}(u, v) + 1$ (resp. $\tilde{y}(u) + \tilde{y}(v) = y(u) + y(v) - \ell_v + \ell_t \geq \bar{c}(u, v)$).

For case (iii), if $(u, v)$ is a forward (resp. backward) edge then $u \in B$, $v \in A$ (resp $u \in A$, $v \in B$). The updated dual weights are $\tilde{y}(u) = y(u) + \ell_t - \ell_u$ (resp. $\tilde{y}(u) = y(u) - \ell_t + \ell_u$) and $\tilde{y}(v) = y(v)$. Note that from (12) and (iii), $\ell_t - \ell_u \leq \ell_v - \ell_u \leq s(u, v)$ (resp. $\ell_u - \ell_t \geq \ell_u - \ell_v \geq -s(u, v)$), we have $\tilde{y}(u) + \tilde{y}(v) = y(u) + \ell_t - \ell_u + y(v) \leq y(u) + y(v) + s(u, v) = \bar{c}(u, v) + 1$ (resp. $\tilde{y}(u) + \tilde{y}(v) = y(u) + \ell_u - \ell_t + y(v) \geq y(u) + y(v) - s(u, v) = \bar{c}(u, v)$). The last equality follows from the definition of slack for forward (resp. backward) edges.

Finally, the augment procedure may introduce new edges into the residual network. Any such forward (resp. backward) edge will not violate the 1-feasibility conditions (3) (resp. (4)). An augmentation will create a new forward edge (resp. backward edge) $(u, v)$ only if flow was pushed along the backward edge (resp. forward edge) $(v, u)$. Since $(v, u)$ is on the augmenting path, $(v, u)$ is an admissible backward (resp. forward) edge, we have $y(a) + y(b) - \bar{c}(u, v) = 0$ (resp. $\bar{c}(u, v) + 1 - y(u) - y(v) = 0$). Therefore, the slack on the forward (resp. backward) edge $(u, v)$ is $s(u, v) = \bar{c}(u, v) + 1 - y(u) - y(v) = 1$ (resp. $s(u, v) = y(u) + y(v) - \bar{c}(u, v) = 1$) implying that the newly added forward (resp. backward) edge satisfies (3) (resp. (4)).

**Proof of (I2):** Consider the shortest path $P'$ from $s$ to $t$ in the augmented residual network. Let $b$ be the free supply node after $s$ and $a$ be the free demand node before $t$ along $P'$. we show that, after the dual updates conducted by Hungarian search, the path $P$ from $b$ to $a$ is an admissible augmenting path. For any edge $(u, v)$ on $P$, by construction $\ell_u \leq \ell_t$ and $\ell_v \leq \ell_t$. Repeating the calculations of case (i) of the proof of (I1), the updated feasibility condition is $\tilde{y}(u) + \tilde{y}(v) = y(u) + y(v) + \ell_v - \ell_u$ (resp. $\tilde{y}(u) + \tilde{y}(v) = y(u) + y(v) - \ell_v + \ell_u$). Note that for edges on the shortest path $P$, (12) holds with equality and so, $\tilde{y}(u) + \tilde{y}(v) = y(v) + y(v) + s(u, v) = \bar{c}(u, v) + 1$ (resp. $\tilde{y}(u) + \tilde{y}(v) = y(v) + y(v) - s(u, v) = \bar{c}(u, v)$), i.e., the path $P$ is an admissible augmenting path. The partial DFS step conducts a search from every free supply vertex including $b$. This will lead to the discovery of at least one augmenting path in the admissible graph.

Next, we show that at the end of partial DFS step, there is no augmenting path in the admissible graph. The partial DFS step maintains a graph $\mathcal{A}$ that is initialized to the admissible graph. After each DFS execution, every edge visited by the DFS except those on the augmenting path $P$ are deleted from $\mathcal{A}$. If no augmenting path is found, every edge and vertex visited by the DFS is deleted from $\mathcal{A}$. The partial DFS step ends when $\mathcal{A}$ does not have any free supply vertices remaining. Note that every

vertex removed from $\mathcal{A}$ is a vertex from which the DFS search backtracked. Similar to Lemma 2.3 (in Gabow and Tarjan SIAM J. Comput. '89), it can be shown that there is no directed cycle consisting of admissible cycles and so, there is no path of admissible edges from any such vertex removed from $\mathcal{A}$ to a free demand node. Since, at the end of the partial DFS step, every free supply vertex is deleted from $\mathcal{A}$, there are no admissible paths from any free supply vertex to a free demand vertex in the admissible graph.