[Reviews · NeurIPS 2019]

Reviewer 1



The paper provides a nice link to more classical literature on graph theoretic algorithms for maximum flows for bipartite matching and adapts a classic algorithm to the specific problem at hand. The detailed analysis will be of interest to algorithmic researchers, and experiments seem to show a practical advantage as well, which is quite exciting. While I did not check the mathematics line-by-line, the treatment seems careful.

Reviewer 2



Edit: I thank the authors for answering my questions in their response. I have raised my rating to a full accept. This is a nice paper! ========= =1. I checked all proofs, and they seem correct to me. (A few minor corrections below). However, the one thing I miss is that the invariants (I1) and (I2) in L259-262—although critical to the analysis—are never proved. These are not immediately obvious to me, please explain the proof in the rebuttal and paper. (They are also stated in L259-261, but aren’t proved there either.) =2. Parallelization is not discussed. A key feature of the Sink alg that makes it very scalable and thus led to its widespread use, is its parallelizability (it requires only matrix-vector mult). It seems to me that the proposed alg is intrinsically not amenable to parallelization. Indeed, the alg proceeds in phases, and in each phase it does Djikstra’s alg, and then sequential iterations of DFS. Each of these parts is hard to/cannot be parallelized well. Please comment on this in the rebuttal and paper. =3. I have concerns with the empirical comparisons and the strong claim in the abstract that the “empirical results suggest our algorithm significantly outperforms existing approaches in execution time”. This is for several reasons: =3a. Note the reported runtimes are shockingly large for such small problem sizes. Indeed, runtimes are in the 10s of seconds or few minutes, for only n=784 (tiny for Sink). This is undoubtedly because the precision demanded is *extremely* high: delta/C roughly between 0.2/1000=2E-5 and 0.02/1000=2E-6. (See below for why the ratio delta/C is the relevant quantity.) The runtime should be reported for delta/C in a range starting from a much more moderate precision e.g. 1e-2. I wonder if/suspect Sink is faster in this moderate-error regime? This is important since in practice, many ML applications only compute OT to a moderate precision. (Obviously this last sentence is somewhat generalizing since error requirements vary by application, but it is my understanding this is often true in practice.) Moreover, the implementation you are using appears to be extremely numerically unstable for such high precision. Indeed, it requires flops with #s of size exp(-4*log(n)*C/delta). Even for the lowest precision you use, delta/C=1e-5, this means numbers of size roughly exp(-1,000,000) which is absurd numerically. Such numerical errors likely worsen Sink’s error, and moreover may also make Sink run much slower as matrix scaling becomes a much harder problem when the matrix to scale has more zero entries (note that exp(-1,000,000) will just be 0 on a computer), see e.g. [Linail et al. “A deterministic strongly polynomial algorithm…”]. There are various tricks used pervasively in implementations of Sink to deal with such numerical issues, see e.g. [Peyre-Cuturi’s book Ch 4.4.] =3b. Scalability in the problem size n should also be investigated. Please plot runtimes for varied n, when delta/C is fixed. I suspect the Sink alg will be able to scale to larger problems than the proposed alg, in part since Sink is parallelizable while the proposed alg is not (see 2 above). =3c. It should be noted that Sink iterations can be implemented significantly faster, in o(n^2) time, when the cost matrix has certain structures that often occur in practice. This occurs, for instance, even in the real-world dataset used in your experiments—although this is not implemented in your paper. Indeed, there, datapoints lie on a uniform grid (pixels in an image), so Sink iterations can be computed in \tilde{O}(n) time rather than n^2 using FFT (see e.g. Peyre-Cuturi book, Ch 4.3). Moreover, even if the points weren’t on a grid, since you are using squared Euclidean distance, the Sink iterations can still be computed to provably sufficient accuracy in \tilde{O}(n) time (see “Massively scalable Sinkhorn distances via the Nystrom method”). =I also find several plots misleading: == For additive approximations, it is only the (scale-invariant) ratio delta/C that is meaningful for understanding the error, *not* delta by itself. All plots should be rescaled so that C is fixed at 1, and only delta is changing. ==I find Figure 2c misleading since you are potting actual error in terms of the parameter delta, yet are changing the error favorably for your alg (L330-336). I would remove Fig 2c. (If you still want to incorporate this information, perhaps change the x-axes of Fig 2a,b to be “error” rather than “delta”.) EXPOSITION =The primal and dual LP formulations should be written somewhere. (Especially since the paper uses a slightly unconventional unbalanced version of OT.) This would give simple, natural explanations of condition (C) as complementary slackness (L178), would clarify what is meant by “dual feasibility conditions” (L169), etc. Also I think the proof of Lemma 2.1 can be significantly shortened using duality/complementary slackness of the LP. =Some re-organization would help the reader. Consider making S1.1 (cost scaling) its own section. Also consider dividing what is currently S2.1 (the algorithm) at L259, to separate the alg’s description and its analysis. =For NeurIPS, it is highly encouraged to include pseudocode of the full alg somewhere. Perhaps put in the Supplement if lack of space. =The proofs would benefit from a re-organization pass through, as several times things are proved about your alg in weird orders and it takes the reader some time to figure out what part of the proof you are referencing. E.g., condition (C) is proved in L263-273 in paragraph form (as opposed to in a lemma), then later this is used in App L36 without reference--consider putting this in a lemma for easier referencing =L259-262: In the invariants, integrality should be mentioned (for dual variables y, slacks, etc.)-----since although simple, this is used in the following proofs repeatedly, and was the whole point of the cost-scaling in S1.1 MINOR (FIXABLE) ERRATA IN PROOFS =L193: typo: eq (7) missing delta’/2 factor in second summand =L269-270: “Since the claim is true initially, it is sufficient to argue that…” It must also be argued that no non-free demand vertex can later become free. This is not hard, but should be mentioned. =App L9: “exactly one” is not true. It can be >1, but always come in telescoping pairs, so OK. =App L44: sum_P Phi(P) is missing r_P factors =App L46: “there are at most \mathcal{U} augmenting paths” is not the right justification. It is because \sum_{P in \mathcal{P}} r_P — i.e. the total pushed flow, summed over all augmenting paths — is at most \mathcal{U}. Indeed this implies \sum_{P \in mathcal{P}} r_P \Phi(P) <= \mathcal{U} * (2) as desired =App L47: first equality should be inequality, see (1) in main text MINOR COMMENTS =L71-72: are there runtimes independent of the dimension of the points? 
=L129 typo: with for =L138 typo: comma should be period =L166 typo: any -> a =L178 typo: y(v) -> y(a) =L181: replace “such that …. is 0” just with “satisfying (C)”. =L190: typo:” missing a =L233: mention briefly that slacks all non-negative (by 1-feasibility eqs (3),(4)), which is Djikstra’s alg can be used =L224:”several” -> at least one =L242-248: mention briefly that this must be implemented with DFS not BFS =Inconsistency in the notation G_sigma. Sometimes with vector over it. =References missing capitalization, e.g. GAN, Wasserstein, Hungarian, Gaussian, Sinkhorn, etc =App L14: typo: comma after because =App L26: consider changing notation “wr” to single letter =App eq (3): align parentheses size

Reviewer 3



The authors propose an approximation algorithm for computing the optimal transport that improves the time complexity of the best known comparable algorithm by a polylog factor and is efficient in practice. The paper provides a comprehensive review of the related work and puts the result well into context. The paper is generally well-written. However, comprehending the technical part requires familiarity with graph algorithms for flow and assignment problems and could probably be clarified. To satisfy the space limit, some proofs might be moved to the appendix. The result is obtained by a combination of classical scaling and primal-dual optimization techniques. The strength of the paper is its theoretical analysis. The approach allows for efficient implementations, which is demonstrated by the experimental evaluation. However, the experimental evaluation is the weakest part of the paper and compares the algorithm to the Sinkhorn method only. A comparison to other methods for the same problem would be desirable.

[Author Response · NeurIPS 2019]

**Proof of Invariants (Reviewer 2 Question 1):** We mildly modify the 1-feasibility condition (3) and enforce it only on any edge $(a, b)$ with a flow $\sigma(a, b) < \min\{s_b, d_a\}$, i.e., only on forward edges in the residual network. While this change does not impact the algorithm or its correctness, it significantly simplifies the proof of invariants. In the submitted version, only Lemma 2.1 uses (3) in (L193). We adapt this proof for the new definition of (3). Initially transform $\sigma$ and $\sigma'$ as follows. For any edge $(a, b)$ that does not satisfy (3), its flow $\sigma(a, b)$ is $\min\{s_b, d_a\}$. We reduce $d_a$ and $s_b$ by $\sigma'(a, b)$, reduce the flow on the edge $(a, b)$ in $\sigma'$ to 0, and reduce $\sigma(a, b)$ to $\min\{s_b, d_a\} - \sigma'(a, b)$. Both $\sigma$ and $\sigma'$ continue to be maximum transport plans and this transformation does not change the difference in their costs. Additionally, every edge with a positive flow in $\sigma'$ now satisfies (3) and (L193) continues to hold.

**Proof of (I1):** We show that after the dual updates of a Hungarian search, every forward edge satisfies (3) and every backward edge satisfies (4). From the shortest path property, we have **(E1):** $\ell_u + s(u, v) \geq \ell_v$. There are four possibilities: (i) $\ell_u < \ell_t$ and $\ell_v < \ell_t$, (ii) $\ell_u \geq \ell_t$ and $\ell_v < \ell_t$, (iii) $\ell_u < \ell_t$ and $\ell_v \geq \ell_t$ or (iv) $\ell_u \geq \ell_t$ and $\ell_v \geq \ell_t$. We present the proof for case (i); the other three cases are similar. For case (i), if $(u, v)$ is a forward (resp. backward) edge then $u \in B$, $v \in A$ (resp. $u \in A$, $v \in B$). The updated dual weights $\tilde{y}(u) = y(u) + \ell_t - \ell_u$ (resp. $\tilde{y}(u) = y(u) - \ell_t + \ell_u$) and $\tilde{y}(v) = y(v) - \ell_t + \ell_v$ (resp. $\tilde{y}(v) = y(v) + \ell_t - \ell_v$), and the updated dual weight sum is $\tilde{y}(u) + \tilde{y}(v) = y(u) + y(v) + \ell_v - \ell_u$ (resp. $\tilde{y}(u) + \tilde{y}(v) = y(u) + y(v) - \ell_v + \ell_u$). From (E1), $\tilde{y}(u) + \tilde{y}(v) \leq y(u) + y(v) + s(u, v) = \lfloor 2c(u,v)/\delta' \rfloor + 1$ (resp. $\tilde{y}(u) + \tilde{y}(v) \geq y(u) + y(v) - s(u, v) = \lfloor 2c(u,v)/\delta' \rfloor$). The last equality follows from the definition of slack for a forward (resp. backward) edge. Note that the augment procedure may introduce new edges into the residual network. Any such new forward (resp. backward) edge will have a slack of 1 because the corresponding backward (resp. forward) edge is admissible implying they satisfy (3) (resp. (4)).

**Proof of (I2):** We show that, after the dual updates are conducted by Hungarian search, the shortest path $P$ from $s$ to $t$ (ignoring vertices $s$ and $t$) is an admissible augmenting path between free vertices $b$ and $a$. For any edge $(u, v)$ on $P$, by construction $\ell_u \leq \ell_t$ and $\ell_v \leq \ell_t$. Repeating the calculations of case (i) of the proof of (I1), the updated dual weight sum is $\tilde{y}(u) + \tilde{y}(v) = y(u) + y(v) + \ell_v - \ell_u$ (resp. $\tilde{y}(u) + \tilde{y}(v) = y(u) + y(v) - \ell_v + \ell_u$). Note that for edges on the shortest path $P$, (E1) holds with equality and so, $\tilde{y}(u) + \tilde{y}(v) = y(u) + y(v) + s(u, v) = \lfloor 2c(u,v)/\delta' \rfloor + 1$ (resp. $\tilde{y}(u) + \tilde{y}(v) = y(v) + y(v) + s(u, v) = \lfloor 2c(u,v)/\delta' \rfloor$), i.e., the path $P$ is an admissible augmenting path. The partial DFS step conducts a search from every free supply vertex including $b$. This will lead to the discovery of at least one augmenting path in the admissible graph. We show that at the end of partial DFS step, there is no augmenting path in the admissible graph. Note that any vertex $v$ removed from $\mathcal{A}$ is a vertex from which the DFS search backtracked. Due to the fact that Hungarian search does not create a cycle in the admissible graph [Lemma 2.3, Gabow Tarjan SICOMP'89], we can conclude that there is no admissible path from $v$ to any free demand node. So the deleted vertex and edge could not have participated in an augmenting path of admissible edges. At the end of partial DFS step, every free supply vertex is deleted from $\mathcal{A}$, so there are no admissible augmenting paths from any free supply vertex.

**Scalability and Parallel Implementations (R2 Q2 & Q3c):** As shown in [Sharathkumar, Agarwal SODA 2012 Section 3.1, 3.2], Hungarian search and partial DFS can be implemented in $\mathcal{O}(n\Phi(n))$ time, where $\Phi(n)$ is the query/update time of a dynamic weighted nearest-neighbor (DNN) data structure. Many distances including squared Euclidean distance admits DNN with poly-logarithmic time search/update operations. We achieve $\tilde{\mathcal{O}}(n)$ execution time for these distances. See also [Agarwal Sharathkumar STOC 14, Section 4 (i)–(iii)] for a relative $\varepsilon$-approximation algorithm using the approximate nearest neighbor (ANN) data structure. Design of ANN-based near-linear time additive approximations and their practical implementation will be stated as potential open questions for the future.

An alternate implementation of Gabow-Tarjan's algorithm [Lahn, Raghvendra SODA 2019, Section 2.1] may be easier to parallelize. This implementation does not require Hungarian Search but only an iterative execution of partial DFS from each free vertex in the admissible graph. Paths in an admissible graph are of length $\mathcal{O}(C/\delta)$ (similar to Lemma 2.4 proof) which may aid in the execution of DFS from each free vertex in parallel. We will pose parallel implementation of our algorithm as an important open question. In our paper and experiments, we focused on sequential execution only.

**Experiments (R2 Q3a, Q3b):** Note that the value of $C$ for all Sinkhorn comparisons (Figure 2) in our paper was fixed at 7.112 (squared Euclidean costs scaled by the median cost), so $\delta/C$ was in the range of $[0.0035, 0.028]$ (includes moderate values). When comparing with other algorithms, we understand that the actual execution time may vary based on the choice of programming language, implementation details and the available resources. Therefore, we give a comparison (right) of the number of iterations taken by our algorithm with those in Table 1 for moderate values of $\delta/C$ (with a setup similar to Figure 2). For Sinkhorn, APDAGD, and our algorithm, each iteration takes $\Theta(n^2)$ time. For the Greenkhorn algorithm, the number of iterations is given by the total row/column updates divided by $n$. In the cases we tested, our algorithm performs fewer iterations on average. We saw similar results for $n \in [500, 2500], \delta/C = 0.05$ on synthetic data. Note that the time taken for augmentations in our algorithm was negligible for our tests.



[Meta-Review · NeurIPS 2019]

We thank the authors for their submission. The quality of their work was appreciated by all reviewers. I feel a bit more uneasy about the quality of their experimental validation. Several choices are surprising. For instance using matlab's linprog is borderline dishonnest given the importance of comparing oneself with a proper network flow solver, and not a generic off-the-shelf LP solver. Comparisons with the sinkhorn algorithm also denote a lack of familiarity with these solvers and their implementation (as highlighted by R2 some of the computational times initially reported are not realistic). Claims about lack of parallelization by R2 were also left unanswered. For all these reasons, and for the interest of the paper, I would strongly recommend that the authors spend significant time improving the experimental section for the camera ready. A Neurips requires this to have visibility (as opposed to other more TCS conferences)